# The Susceptibility Trends of Respiratory and Enteric Porcine Pathogens to Last-Resource Antimicrobials

**DOI:** 10.3390/antibiotics12111575

**Published:** 2023-10-28

**Authors:** Anna Vilaró, Elena Novell, Vicens Enrique-Tarancon, Jordi Baliellas, Lourdes Migura-García, Lorenzo Fraile

**Affiliations:** 1Grup de Sanejament Porcí, 25192 Lleida, Spain; micro@gsplleida.com (A.V.); elena@gsplleida.net (E.N.); vicens@gsplleida.net (V.E.-T.); jordi@gsplleida.net (J.B.); 2Unitat Mixta d’Investigació IRTA-UAB en Sanitat Animal, Centre de Recerca en Sanitat Animal (CReSA), 08193 Bellaterra, Spain; lourdes.migura@irta.cat; 3IRTA, Programa de Sanitat Animal, Centre de Recerca en Sanitat Animal (CReSA), 08193 Bellaterra, Spain; 4Departament de Ciència Animal, ETSEA, University of Lleida-Agrotecnio, 25198 Lleida, Spain

**Keywords:** trend analysis, antimicrobial susceptibility, porcine pathogens

## Abstract

Monitoring the antimicrobial susceptibility of last-resource antimicrobials for veterinary pathogens is urgently needed from a one-health perspective. The objective of this study was to analyze the antimicrobial susceptibility trends of Spanish porcine bacteria to quinolones, cephalosporins, and polymyxins. Isolates of *Actinobacillus pleuropneumoniae*, *Pasteurella multocida*, and *Escherichia coli* were isolated from sick pigs from 2019 to 2022. An antimicrobial susceptibility test was determined based on the minimal inhibitory concentration (MIC) following an internationally accepted methodology. The MIC categorization was based on distributing the range of MIC values in four categories, with category one being the most susceptible (lowest MIC value) and category four the least susceptible (highest MIC value). Moreover, clinical susceptibility (susceptible/non-susceptible) was also determined according to the CLSI and EUCAST clinical breakpoints. A logistic and multinomial logistic regression model was used to analyze the susceptibility data for dichotomized and categorized MIC data, respectively, for any pair of antimicrobial/microorganism. In general terms, the antimicrobial susceptibility of pig bacteria to these antimicrobials remained stable or increased in the last four years in Spain. In the case of *A. pleuropneumoniae* and quinolones, a significant temporal trend was observed where isolates from 2020 had significantly increased odds of being more susceptible than isolates from 2019. In the case of *E. coli* and polymyxins, a significant temporal trend was observed where isolates from 2020 and 2021 had significantly increased odds of being more susceptible than isolates from 2019 and 2020, respectively. Finally, significant odds of being less susceptible were only observed for cephalosporins and *E. coli* for 2020 versus 2019, stagnating for the rest of study period. These results provide sound data on critically important antimicrobials in swine medicine.

## 1. Introduction

Antimicrobial resistance (AMR) threatens the successful treatment of bacterial infections, not only in humans but also in animal health [1,2]. The use of antimicrobials (ABs) in humans and animals is a driver in the increase in AMR in bacterial populations, even following guidelines for the prudent use of ABs [3,4]. This risk significantly increases with the misuse of these drugs, but other factors are also involved in the increase in AMR [5]. Moreover, the AMR reservoir of bacteria from livestock has been increasingly investigated for its potential to transfer AMR to humans via direct contact, the environment, and contaminated food [6,7,8]. Nevertheless, the extent of this transmission remains uncertain due to the enormous complexity of the AMR epidemiology involving animals, the environment, and humans [9,10,11]. Nevertheless, policymakers in the European Union (EU) have developed legislation to monitor and regulate antimicrobial use in animals with the goal to decrease AMR burden in humans in the long run [12,13]. However, the global effect of these actions regarding the reduction of AMR at the human–animal–environment interface is still under investigation, and very few scientific studies have shown encouraging results, limited to some antimicrobials such as colistin [14,15,16]. This could be since antimicrobial use (AMU) is one key driver for AMR, but other socio-economic factors should be also considered in AMR epidemiology, as recently assessed [17]. These studies highlight the relevance of tackling AMR using a one-health approach, where control measures should be addressed to humans, animals, and the environment. On the other hand, this long-term reduction of AB consumption in veterinary medicine could seriously hamper the care of animals and generate severe welfare issues if animals are not treated with the right antimicrobial when it is really needed.

The current EU legislation regarding antimicrobials [12] has focused special attention to restrict as much as possible the use of last-resource antimicrobials (third- and fourth-generation cephalosporins, polymyxins, and quinolones) in animals, following the recommendations addressed by the European Medicine Agency in 2019 [18]. Thus, these last-resource ABs can only be used when no other options belonging to less risky categories (C and D) for AMR are available to treat animals [18]. However, up to date, most of the long-term surveillance data available are only from healthy animals that may not reflect the situation in veterinary bacterial pathogens [19]. Thus, the European Food Safety Authority (EFSA) coordinates a mandatory active monitoring of AMR in zoonotic (*Salmonella species* and *Campylobacter* spp.), indicator bacteria (*Escherichia coli*), and extended-spectrum-cephalosporin-resistant and carbapenemase-producing *E. coli* from healthy food-producing animals (cattle, poultry, pigs) at slaughter and in meat, following European directives [20,21]. On the other hand, a coordinated and harmonized strategy for AMR monitoring in diseased animals has just started at the European level [22] to fill the gap for AMR data in pathogens from diseased animals. Thus, updated information will be generated to guide antimicrobial stewardship initiatives such as treatment guidelines, and to guide policymakers in regulating veterinary antimicrobial use [23].

The use of antimicrobials for therapeutic or metaphylactic purposes in pigs may be necessary to control the relevant pathogens involved in respiratory and enteric disorders, contributing to most of the pig antimicrobial consumption [24,25,26]. Thus, porcine respiratory disease complex (PRDC) and post-weaning diarrhea (PWD) are some of the most challenging diseases affecting the pig industry worldwide [27,28]. PRDC is a syndrome that results from a combination of infectious (bacteria and viruses) and non-infectious factors. Moreover, *Actinobacillus pleuropneumoniae* (APP), *Pasteurella multocida*, *Mycoplasma hyopneumoniae*, and *Bordetella bronchiseptica* are the most common bacterial agents involved [29]. On the other hand, *Escherichia coli* is the main causative agent of PWD, affecting piglets after weaning. PWD is characterized by profuse diarrhea, dehydration, significant mortality, and loss of body weight in surviving pigs [30,31,32]. When clinical signs appear, the prescription of antimicrobials is, in many cases, the only solution to control the spread of the PRDC and PWD within the herd [24,25,32,33,34]. Thus, it may be necessary to use last-resource antimicrobials if no other option of less risky categories for AMR is available according to an antimicrobial stewardship program [3,4,5,6,7,8,9,10,11,12,13,14,15,16,17,18,19,20,21,22,23,24,25,26,27,28,29,30,31,32,33,34,35]. It must be highlighted that, during the last four years, the sales of last-resource antimicrobials in European livestock were between 0.2 and 2.8% of the total sales of antimicrobials [36], suggesting that bacterial populations are hardly exposed to this family of drugs across Europe.

An important aspect of dealing with the AMR crisis is surveillance [37], which provides susceptibility data, allowing for more effective action when necessary. Another goal of AMR surveillance is to analyze the temporal trends of AMR patterns for the early warning of potential threats, and to decipher the impact of policies in animals regarding the use of antimicrobials in the long term. Unfortunately, there is scarce knowledge on the antimicrobial susceptibility profiles of veterinary bacterial pathogens in Europe due to a lack of coordinated strategies between member states [23]. The objective of this study was to describe and analyze the temporal trends during the last four years of last-resource antimicrobials in Spanish porcine pathogens as a suitable model for other countries, considering the low consumption of these drugs in Spain compared with the total antimicrobial consumption (3–4,1%) and the consistent decrease in the total antimicrobial use in livestock [36].

## 2. Results

### 2.1. Bacteria Isolation

From January 2019 to 2022, 1650 samples were received from isowean, wean-to-finish, and fattening pigs suffering from clinical respiratory disease associated with PRDC. Additionally, 3646 samples were received from sow, isowean, and wean-to-finish farms suffering clinical signs compatible with PWD. Only one isolate was included per farm across the study to avoid redundancy and the over-representation of bacterial clones. In the case of sow farms, the samples were obtained from their nursery facility. Bacterial isolation for respiratory pathogens (*A. pleuropneumoniae*, *P. multocida*, *and B. bronchiseptica*) was successful in 80% (1319/1650) of the cases. Furthermore, in 20% of the samples, more than one bacterial species was isolated. The bacterial isolation of *E. coli* was successful in 79.3% (2892/3646) of the samples associated with enteric disorders. Finally, in 5% of the enteric samples, it was possible to isolate more than one bacterial species, generally *Salmonella species*. The number of isolates is detailed in Table 1. Thus, for *A. pleuropneumoniae*, *E. coli*, and *P. multocida*, there were at least 100 isolates isolated each year and, therefore, they were included in the statistical analysis, whereas there were only 24–53 and 18–52 isolates per year for the *Bordetella bronchiseptica* and *Salmonella species*, respectively, during the study period.

### 2.2. Distribution of MIC per Antimicrobial and Microorganism across the Years

The MIC distributions (MIC range, MIC_50_ and MIC_90_) are shown in Table 2, Table 3 and Table 4 for *A. pleuropneumoniae*, *P. multocida*, and *E. coli* to quinolones (enrofloxacin and marbofloxacin), cephalosporins (ceftiofur and cefquinome), and polymyxins (colistin) during the study period.

In the case of quinolones, there were isolates with low and extremely high MIC values in the same distribution (MIC range of 0.03–4) for all the bacterial pathogens, but the MIC_90_ was lower for respiratory pathogens (*A. pleuropneumoniae* and *P. multocida*) than for digestive ones (*E. coli*) across the study period. Moreover, the MIC_90_ remained stable across the study period for all the bacterial pathogens, or slightly decreased in the case of APP (Table 2).

In the case of cephalosporins, the MIC range for respiratory pathogens (0.06–1) was smaller than for digestive ones (0.06–8) (Table 3). Thus, there were *E. coli* isolates with low and extremely high MIC values in the same distribution. Moreover, the MIC_90_ was also lower for respiratory pathogens (*A. pleuropneumoniae* and *P. multocida*) than for digestive ones (*E. coli*). In both cases, the MIC_90_ remained stable across the study period (Table 3).

In the case of polymyxins, the MIC_90_ sustainably decreased from 2019 to 2022, but the MIC range remained similar during the study period (Table 4).

### 2.3. Logistic and Multinominal Model for Quinolones

After statistical analysis using dichotomized (susceptible/non-susceptible) and categorized MIC data (Figure 1 and Figure 2), non-significant temporal trends were observed for susceptibility to enrofloxacin in *E. coli* and *P. multocida* (*p* > 0.05). Contrarily, for *A. pleuropneumoniae*, a significant temporal trend (*p* = 0.002) was detected for this antimicrobial. The isolates from 2020 had significantly increased odds of being more susceptible to enrofloxacin than isolates from 2019, comparing MIC category 1 versus 3 and 1 versus 4. Moreover, isolates from 2020 (Table 5 and Figure 2) also had increased odds of being more susceptible than isolates from 2019 using dichotomized MIC data (susceptible/resistant, *p* = 0.0002).

The number of categories for multinominal analysis was based on distributing the range of MIC values in four categories (from one to four), equally distributed, that include two MIC values per category, with category one being the most susceptible (lowest MIC value) and category four the least susceptible (highest MIC value).

In the case of *P. multocida* (Figure 2 and Figure 3), no temporal trend for susceptibility to marbofloxacin was observed (*p* > 0.05). However, in the case of *A. pleuropneumoniae*, a significant temporal trend (*p* < 0.0001) was detected, where isolates from 2020 had significantly increased odds of being more susceptible than isolates from 2019, comparing MIC category 1 versus 3 and 1 versus 4 (Table 6). Thus, isolates from 2021 had significantly decreased odds of being more susceptible than isolates from 2020, comparing MIC category 1 versus 2. However, when using the dichotomized MIC data between 2020 and 2019, a significant temporal trend was observed for this bacteria–drug combination (Table 6 and Figure 2). Finally, in the case of *E. coli* and marbofloxacin, a significant temporal trend (*p* < 0.0001) was also observed, where isolates from 2020 had significantly increased odds of being more susceptible than isolates from 2019, comparing MIC category 1 versus 2, but they had significantly decreased odds of being more susceptible when comparing MIC category 1 versus 3 between these years (Table 7). Interestingly, no significant trend was observed using the dichotomized data (susceptible/non-susceptible) for this drug/microorganism combination (Table 7 and Figure 2).

The number of categories for multinominal analysis was based on distributing the range of MIC values in four categories (from one to four), equally distributed, that include two MIC values per category, with category one being the most susceptible (lowest MIC value) and category four the least susceptible (highest MIC value).

### 2.4. Logistic and Multinominal Model for Third- and Fourth-Generation Cephalosporins

The multinominal regression analysis for *E. coli* identified significant annual variation in susceptibility to ceftiofur (Figure 4). Thus, the *E. coli* isolates from 2020 had significantly decreased odds of being more susceptible than isolates from 2019 when comparing all the MIC categories (1 versus 2, 1 versus 3, and 1 versus 4) (Table 8). However, the *E. coli* isolates from 2021 had significantly increased odds of being more susceptible than isolates from 2020 when comparing MIC category 1 versus 4. Using the dichotomized data, the *E. coli* isolates from 2020 also had significant odds of being less susceptible than isolates from 2019 (Table 8 and Figure 5). In the case of *E. coli* and cefquinome (fourth-generation cephalosporin), no significant temporal trend in antimicrobial susceptibility (*p* > 0.05) was detected using the multinominal regression analysis, (Figure 6) whereas the dichotomized analyses showed that the *E. coli* isolates from 2020 had significantly decreased odds (0.70–(0.52–0.96)) of being more susceptible than the isolates from 2019 (Figure 5).

The number of categories for multinominal analysis was based on distributing the range of MIC values in four categories (from one to four), equally distributed, that include two MIC values for category, with category one being the most susceptible (lowest MIC value) and category four the least susceptible (highest MIC value).

In the case of *A. pleuropneumoniae*, the percentage of isolates belonging to category 1 for ceftiofur was close to 100% across the study period without observing any temporal trend (*p* > 0.05), either with the dichotomized or categorized MIC data. In the cases of *P. multocida* and ceftiofur (Figure 4), a significant temporal trend in antimicrobial susceptibility was observed during the study period (*p* < 0.05). Thus, the isolates from 2020 had significantly decreased odds of being more susceptible than the isolates from 2019 when comparing MIC category 1 versus 2 (Figure 4 and Table 9). However, the isolates from 2022 had significantly increased odds of being more susceptible than the isolates from 2021 when comparing MIC category 1 versus 2 (Table 9). Interestingly, non-significant differences were observed using the dichotomized data (susceptible/non-susceptible) for this combination of drug/microorganism (Figure 5).

The number of categories for multinominal analysis was based on distributing the range of MIC values in four categories (from one to four), equally distributed, that include two MIC values per category, with category one being the most susceptible (lowest MIC value) and category four the least susceptible (highest MIC value).

### 2.5. Logistic and Multinominal Model for Polymyxins

Only *E. coli* was tested against this antimicrobial. A significant temporal trend (*p* < 0.0001) was detected, where the isolates from 2020 had significantly increased odds of being more susceptible than the isolates from 2019 when comparing MIC category 1 versus 2 and 3. This same result was also observed with isolates from 2021 versus isolates from 2020, but only when comparing MIC category 1 versus 2 (Figure 7 and Table 10). On the other hand, the isolates from 2022 had significantly decreased odds of being more susceptible than the isolates from 2021 when comparing MIC category 1 versus the rest of categories (Figure 7B and Table 10). However, using the dichotomized MIC data, only the isolates from 2020 had significantly increased odds of being more susceptible than the isolates from 2019 (Figure 8 and Table 10).

The number of categories for multinominal analysis was based on distributing the range of MIC values in four categories (from one to four), equally distributed, that include two MIC values per category, with category one being the most susceptible (lowest MIC value) and category four the least susceptible (highest MIC value).

## 3. Discussion

Antimicrobial susceptibility is usually measured based on the minimum inhibitory concentration (MIC), which is the lowest concentration that stops the in vitro growth of the targeted bacteria using microdilution methods in veterinary laboratories. Modelling MIC values is challenging, since these types of data are interval-censored and ordinal [38,39]. One approach to deal with these data is to dichotomize the MIC values into two categories, resistant (R) and susceptible (S), using established clinical breakpoints or epidemiological cut-off values (ECOFFs), followed by logistic regression [40,41]. However, this is not an ideal approach, since there is a loss of quantitative information from the MIC values when they are dichotomized [38,42]. Another critical point for dichotomizing the MIC values into R and S categories is the existence of accepted clinical breakpoints to obtain comparable results between different studies. In the case of pig respiratory pathogens, there is a reasonable number of internationally accepted clinical breakpoints, but this is not the case for pig enteric pathogens. Moreover, the EUCAST ECOFFs are missing for 45.3% (MIC) and 76.9% (disk diffusion) of bacterial species in the veterinary field [22]. Since we work with clinical cases, it was decided to interpret our MIC results using clinical breakpoints instead of ECOFFs. Therefore, we can monitor the antimicrobial susceptibility pattern for different antibiotics, but we cannot monitor resistance in bacterial populations as suggested by the European Antimicrobial Resistance Surveillance Network for veterinary pathogens (EARS-VET) [22]. Moreover, our study is based on clinical cases (passive collection) whose representativeness of the general animal population is unknown [43]. Considering the limited information available for some antibiotic–microorganism pair, we have extrapolated the clinical breakpoints available for quinolones and cephalosporins and respiratory pathogens [44,45] to enteric ones, and we have used the clinical breakpoint for colistin and *E. coli* from humans [46]. This approach seems reasonable to study the antimicrobial susceptibility temporal trends for all the porcine pathogens, but it has not allowed for directly extrapolating these findings to clinical efficacy in pigs, especially for digestive pathogens. Despite these limitations, we consider that our data provide robust information about the evolution of the antimicrobial susceptibility pattern of the main pig pathogens in Spain during the study period.

The qualitative categorization into S and R, does not allow for the determination of the dynamics of bacterial populations, particularly wild-type populations approaching the clinical breakpoint. This is especially important for cases of decreased susceptibility to antimicrobials associated with punctual mutations, like fluoroquinolones and *E. coli*, where increases in the MIC are associated with chromosomic mutations in the quinolone resistance-determining regions [47]. MIC outcome data could be more appropriately modelled using statistical models other than logistic regression, such as Cox proportional hazards, multinomial logistic, ordinal logistic, linear, and tobit regression models [38,39,40,42,48]. In this case, we have used a multinominal logistic model based on distributing the range of MIC values into four categories (from 1 to 4) that include two MIC values in each category, with category 1 being the most susceptible (lowest MIC value) and 4 the least susceptible (highest MIC value), as suggested by other authors with a similar database for *E. coli* [49]. Finally, the antimicrobial panel was selected to represent commonly used compounds for the treatment of pig diseases in practice [34,35], and not focused on monitoring antimicrobial resistance in surveillance programs. This is a clear limitation of this study since the antimicrobials tested herein were not the same for all the porcine pathogens.

Our data clearly showed a different pattern in the evolution of antimicrobial susceptibility for each combination of drug and microorganism. However, in the cases of fluoroquinolones, marbofloxacin, and enrofloxacin, in combination with *A. pleuropneumoniae*, the proportion of isolates susceptible to each of the antimicrobials was practically the same. There was a similar occurrence for *P. multocida*, indicating that testing one of the fluoroquinolones in these two pathogens would be sufficient to test for this antimicrobial family [35]. Contrarily, data on susceptibility obtained for *E. coli* in combination with ceftiofur could not be extrapolated to cefquinome, as it has also been previously suggested by other authors [50]. This is not surprising, as cefquinome has been reported to not be useful in separating isolates with extended spectrum betalactamases or plasmidic AmpC from cephalosporin-susceptible isolates [51]. These results reinforced that the evolution of antimicrobial susceptibility must be studied in a case-by-case situation, where generalization for drug families and bacteria is not possible, as described previously [35]. Finally, one interesting line of research could be studying the evolution mechanisms shaping the maintenance of antimicrobial resistance in pig pathogens, as carried out by Durao et al. [52], but this is outside the scope of this paper. 

In general terms, the pig pathogens involved in respiratory diseases analyzed herein appeared to remain susceptible or tended to increase their susceptibility to critical antimicrobials over the study period. For *E. coli*, there was also a tendency to increase susceptibility for most antimicrobials, except for ceftiofur, where there was a significant decrease in susceptibility for MIC category 1 from 2019 to 2020. Taken together, the results obtained using the dichotomized versus categorized MIC data were generally similar for all the pairs of drug/microorganism combinations with some exceptions, where the categorized MIC was more sensitive, detecting slight changes in the antimicrobial susceptibility patterns (i.e., cefquinome and marbofloxacin in combination with *E. coli*). Finally, for the combination of colistin with *E. coli*, by using the dichotomized MIC data, a dramatic increase in susceptibility to colistin from 2019 to 2021 was observed, with slight decrease in 2022. This is interesting, since there was a voluntary reduction in the sales of colistin in pig production in Spain from 34.9 mg/PCU to 3 mg/PCU between 2015 and 2018 [36], which could explain these results, but we do not have figures of colistin consumption per farm, and a sound study linking consumption with antimicrobial susceptibility cannot be carried out with our database. Still, by using the dichotomized MIC data, this decrease in susceptibility observed for the year 2022 was not detected, suggesting that categorized MIC data may be more sensible that dichotomized data to detect slight changes in antimicrobial susceptibility patterns. On the other hand, the antimicrobial consumption of livestock has also been reduced by 62.4% from 2014 to 2021 in Spain [36] for drugs focused on respiratory pathogens. Unfortunately, it was not feasible to carry out a study to link this AB consumption in pigs with the antimicrobial susceptibility trend observed for respiratory pathogens because AB consumption is not available at the farm level. It should be a priority to carry out this study using a multivariable model, including the way antimicrobials are used on farms, routes of administration, the duration of antimicrobial use, veterinary control, herd size, and the level of biosecurity and sanitation, as has been carried out recently for human health [17].

In Spain, the antimicrobial susceptibility for last-resource antimicrobials in pig pathogens remained stable or increased in the last four years. These are sound results in terms of preserving the efficacy of critical important antimicrobials and minimizing the burden and spread of resistance from farm to fork.

## 4. Materials and Methods

### 4.1. Clinical Samples

Between January 2019 and December 2022, samples were taken from diseased or recently deceased pigs from farms across Spain showing acute clinical signs of respiratory tract infections or pigs showing diarrhea. None of these animals had been exposed to antimicrobial treatment for at least 15 days prior to sampling. Thus, the sampled animals (1650 animals) were between 3 and 24 weeks old showing overt respiratory symptoms with or without depression and/or hyperthermia (>39.8 °C). For each clinical case, samples of the lungs of two recently deceased pigs (<12 h) were submitted under refrigeration to the laboratory. If no recently dead pigs were suitable for sampling, at least two animals with acute respiratory signs were humanely sacrificed and lung samples were drawn. On the other hand, for piglets showing PWD, the sampled animals (3646 animals) were between 3 and 12 weeks old, showing clinical symptoms of the disease. Fecal swabs were drawn from sick pigs with watery diarrhea or from intestinal content if the animals were humanly euthanized due to their poor clinical conditions. In both cases, the samples were submitted under refrigeration to the laboratory and processed during the following 24 h after collection. Only one isolate was included per farm across the study to avoid redundancy and the over-representation of bacterial clones.

### 4.2. Bacterial Isolation and Identification

Clinical specimens were cultured aseptically onto blood agar (Columbia agar with 5% Sheep blood, 254005 BD, Heidelberg, Germany), chocolate agar (GC II agar with IsoVitaleX, 254060, BD or blood Agar No. 2 Base, 257011, BD Heidelberg, Germany), and MacConkey agar (4016702, Biolife Italiana Srl, Milan, Italy), and incubated at 35–37 °C in aerobic conditions with 5–10% CO_2_ for 24–48 h to address the isolation of respiratory bacterial pathogens. Finally, for the isolation of digestive pathogens, the specimens were cultured aseptically onto blood agar, MacConkey agar, and Xylose-Lysine-Desoxycholate Agar (XLD, CM0469, Oxoid, Thermofisher scientific, Basingstoke, England). The plates were incubated at 35–37 °C in aerobic conditions for 24 h.

The identification of isolates for respiratory pathogens and enteric pathogens was carried out through matrix assisted laser desorption ionization–time of flight (MALDI-TOF Biotyper System, Bruker Daltonics, Bremen, Germany), as previously described (25). The individual isolates were stored at −80 °C in a brain–heart infusion (CM1135, Oxoid) with 30% glycerol (G9012, Sigma-Aldrich). 

### 4.3. Antimicrobial Susceptibility Testing

Antimicrobial susceptibility testing was determined using the minimum inhibitory concentration (MIC) value for each combination of bacterial species and antimicrobial tested. Thus, the MIC was obtained in accordance with the recommendations presented by the Clinical and Laboratory Standards Institute [34,35] in a customized 96-well microtiter plate (Sensititre, Trek diagnostic Systems Inc., East Grinstead, UK) containing a total of 12 and 8 antibiotics/concentrations for respiratory and digestive pathogens, respectively. The antimicrobials tested for swine respiratory pathogens belong to category D [18]: Sulfamethoxazole/trimethoprim, doxycycline, oxytetracycline, and amoxicillin; Category C: Florfenicol, tiamulin, tulathromycin, tildipirosin, and tilmicosin; and category B: Ceftiofur, enrofloxacin, and marbofloxacin. On the other hand, the antimicrobials tested for swine enteric pathogens belong to category D: Sulfamethoxazole/trimethoprim and spectinomycin; Category C: florfenicol, apramycin, gentamycin, neomycin, and amoxicillin/clavulanic acid; and category B: ceftiofur, cefquinome enrofloxacin, marbofloxacin, and colistin. 

The bacteria were thawed, cultured on chocolate agar or blood agar, and incubated at 35–37 °C in aerobiosis (or with 5–10% CO_2_ for APP) for 18–24 h. Three to five colonies were picked and emulsified in demineralized water (or Cation Adjusted Muëller-Hinton Broth (CAMHB) for APP) to obtain a turbidity of 0.5 McFarland standard (Sensititre™ nephelometer V3011). The suspensions were further diluted in CAMHB for *E. coli*, CAMHB or CAMHB with 2.5–5% Lysed Horse Blood for *P. multocida*, and Veterinary Fastidious Medium (VFM) or Mueller Hinton Fastidious broth with Yeast (MHF-Y) for APP to reach a final inoculum concentration of 5 × 10^5^ cfu/mL. Then, the Sensititre panel was reconstituted by adding 100 μL/well of the inoculum. The plates containing *E. coli* isolates were incubated at 35 ± 2 °C for 16–20 h; the *P. multocida* isolates were incubated at 35 ± 2 °C for 18–24 h. In the case of the APP isolates, the plates were covered with a perforated seal and incubated at 35 ± 2 °C with 5–10% CO_2_ for 20–24 h. 

The antibiotic panels were read manually using Sensititre™ Vizion (V2021) and the MIC value was established as the lowest drug concentration inhibiting visible growth. For each isolate tested, a colony count and a purity check were performed following the CLSI and manufacturer recommendations. Moreover, quality control strains were also included. Thus, *Actinobacillus pleuropneumoniae* (ATCC 27090™), *Escherichia coli* (ATCC 25922™), *Streptococcus pneumoniae* (ATCC 49619™), and *Enterococcus faecalis* (ATCC 29212™) were included as quality control following the CLSI recommendations [34,35]. The MICs of the quality control strains had to be within acceptable CLSI ranges to accept the results obtained in the laboratory.

### 4.4. Statistical Methods

All the data analysis was carried out with JMP^®^, Version 13 (SAS Institute Inc., Cary, NC, USA, 1989–2019). Descriptive statistics (MIC range, MIC_50_ and MIC_90_) were performed to summarize the distribution of the isolates within each MIC category. The number of categories was based on distributing the range of MIC values in four categories (from one to four), equally distributed, that include two MIC values per category, with category one being the most susceptible (lowest MIC value) and category four the least susceptible (highest MIC value). The range of concentrations tested were 0.06–8, 0.03–4, and 0.25–32 μg/mL for 3rd- and 4th-generation cephalosporins, quinolones, and polymyxins, respectively. Moreover, clinical susceptibility (susceptible/non-susceptible for each isolate) was determined according to the CLSI clinical breakpoints for APP, *P. multocida*, and *E. coli* for quinolones and cephalosporins, and the EUCAST guidelines for colistin in the case of *E. coli*, respectively [44,45,46] (Table 11).

A logistic (susceptible/non-susceptible for each isolate) and multinomial logistic regression model (four MIC categories) was used to analyze the susceptibility data for the antimicrobials from the years 2019 to 2022, only for those pairs of antimicrobial/microorganisms if at least 100 isolates were available for each year, as recommended by De Jong et al. (2022) [19]. Susceptible/non-susceptible and categorized MIC data (MIC category 1, 2, 3 and 4) were used for the logistic and multinominal logistic regression model, respectively, as dependent variables, and the year as an independent one. Thus, the year of sampling was categorized based on individual years and modelled as a hierarchical indicator variable, where for each year, the preceding year was used as the referent [53]. The final multinomial model was executed with outcome category 1 as the base referent category (the most susceptible one). The model assumptions and goodness of fit were evaluated as appropriate for these models [53]. Thus, the level of significance used to reject a null hypothesis was *p*  ≤  0.05.

## Figures and Tables

**Figure 1 antibiotics-12-01575-f001:**
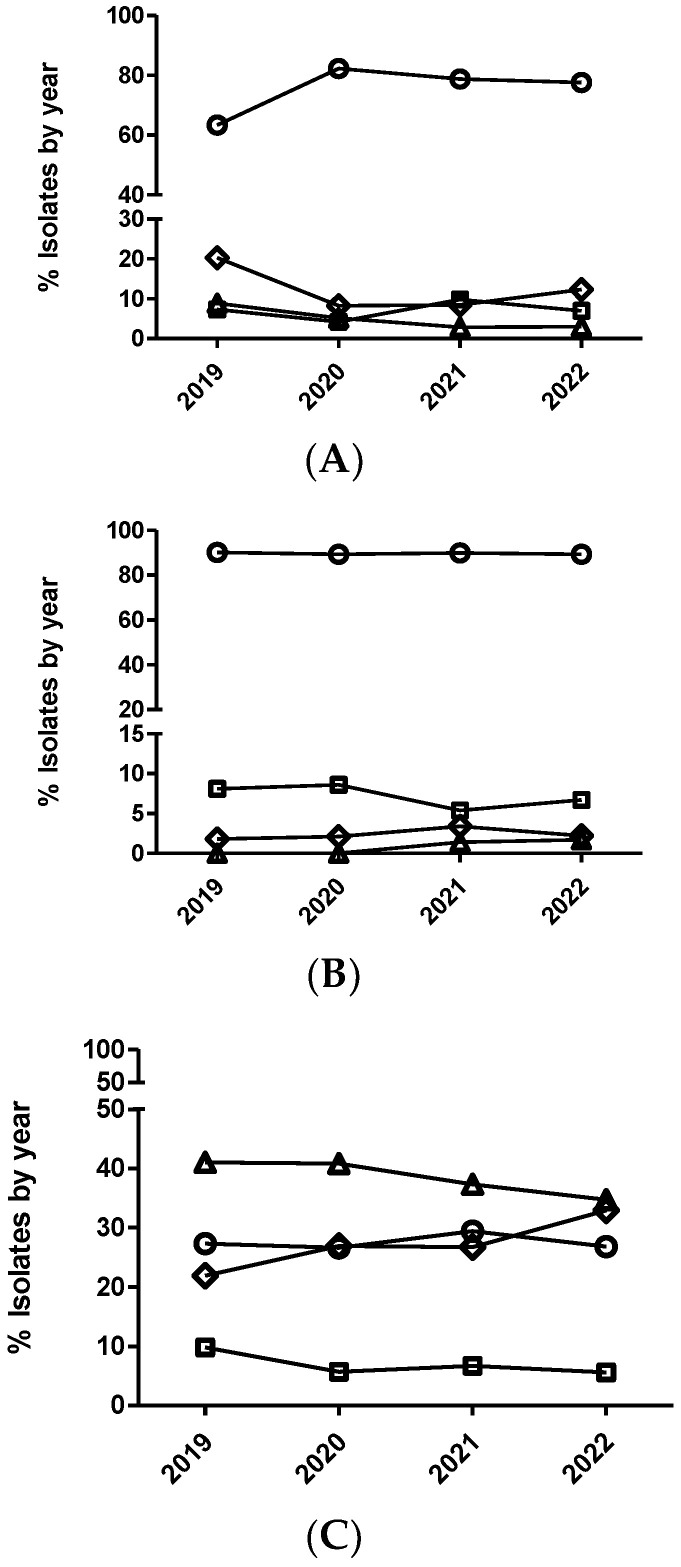
Percentage of *Actinobacillus pleuropneumoniae* (**A**), *Pasteurella multocida* (**B**), and *Escherichia coli* (**C**), and isolates belonging to antimicrobial susceptibility category 1 (circle), category 2 (square), category 3 (diamond), and category 4 (triangle) for enrofloxacin. The number of categories for multinominal analysis was based on distributing the range of MIC values in four categories (from one to four), equally distributed, that include two MIC values per category, with category one being the most susceptible (lowest MIC value) and category four the least susceptible (highest MIC value).

**Figure 2 antibiotics-12-01575-f002:**
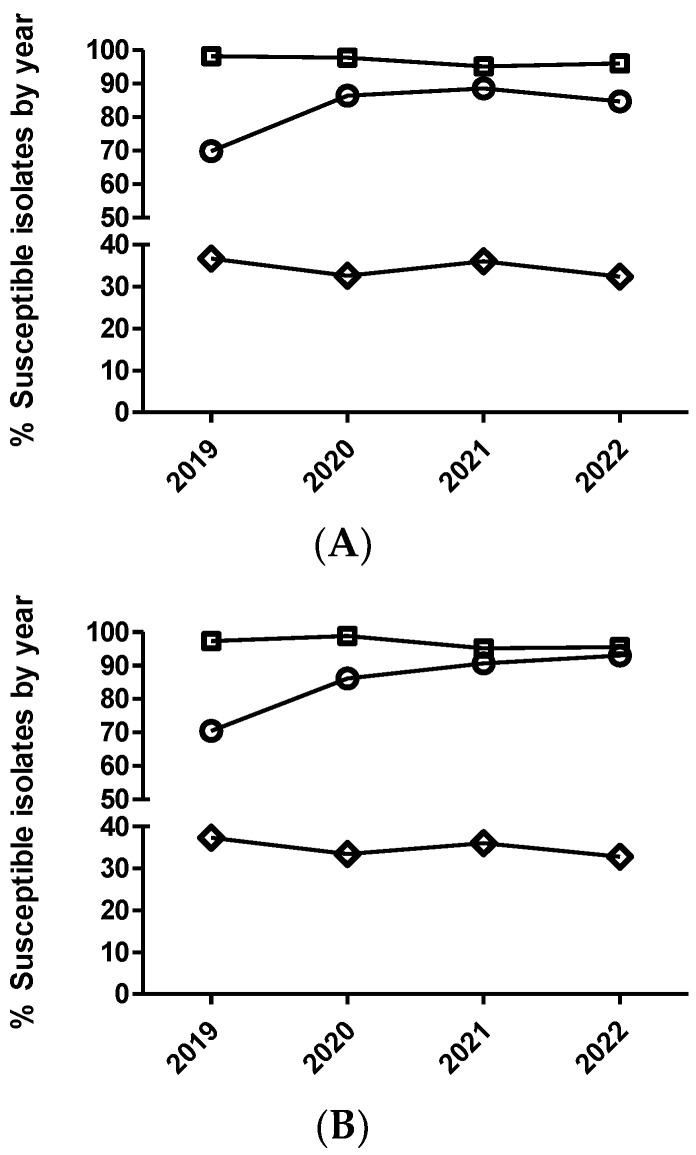
Percentage of susceptible isolates by year for enrofloxacin (**A**) and marbofloxacin (**B**) of *Actinobacillus pleuropneumoniae* (circle), *Pasteurella multocida* (square), and *Escherichia coli* (diamond), using the CLSI and EUCAST clinical breakpoints, as detailed in the Materials and Methods section.

**Figure 3 antibiotics-12-01575-f003:**
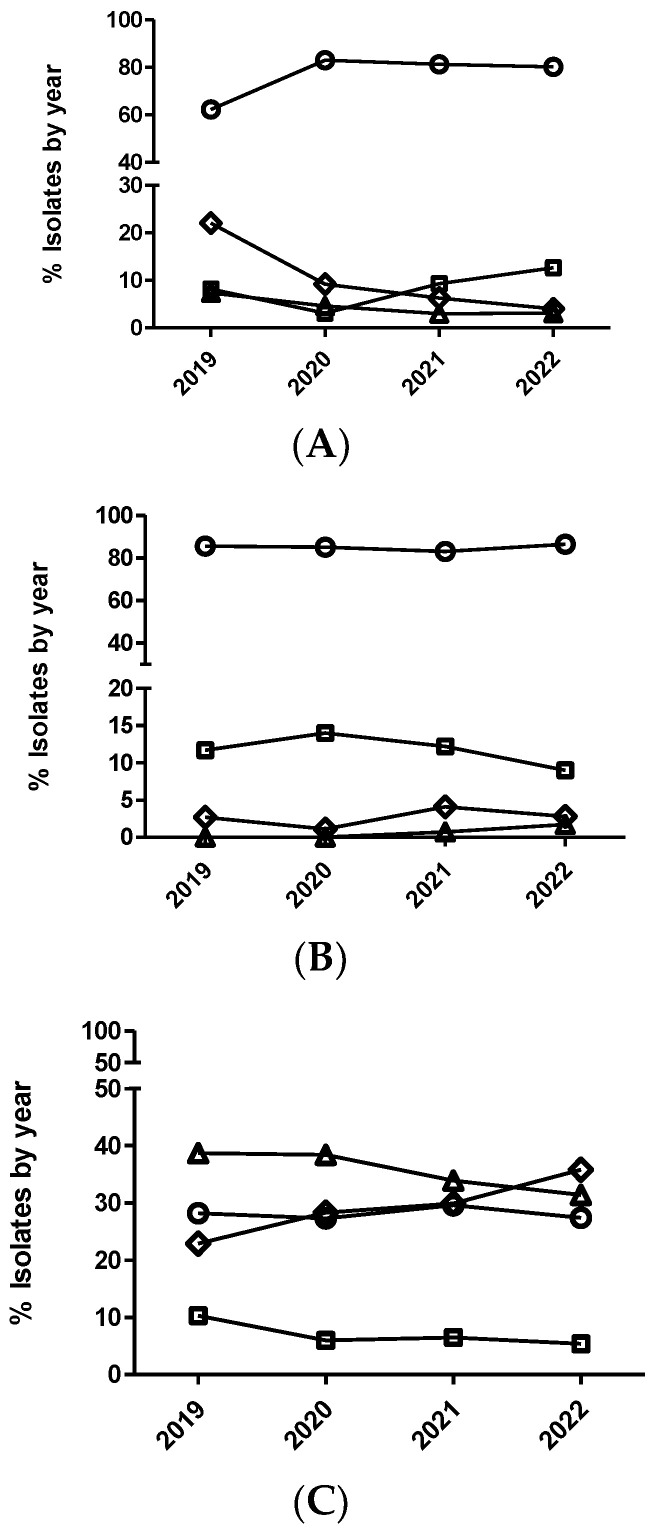
Percentage of *Actinobacillus pleuropneumoniae* (**A**), *Pasteurella multocida* (**B**), and *Escherichia coli* (**C**), and isolates belonging to antimicrobial susceptibility category 1 (circle), category 2 (square), category 3 (diamond), and category 4 (triangle) for marbofloxacin. The number of categories for multinominal analysis was based on distributing the range of MIC values in four categories (from one to four), equally distributed, that include two MIC values per category, with category one being the most susceptible (lowest MIC value) and category four the least susceptible (highest MIC value).

**Figure 4 antibiotics-12-01575-f004:**
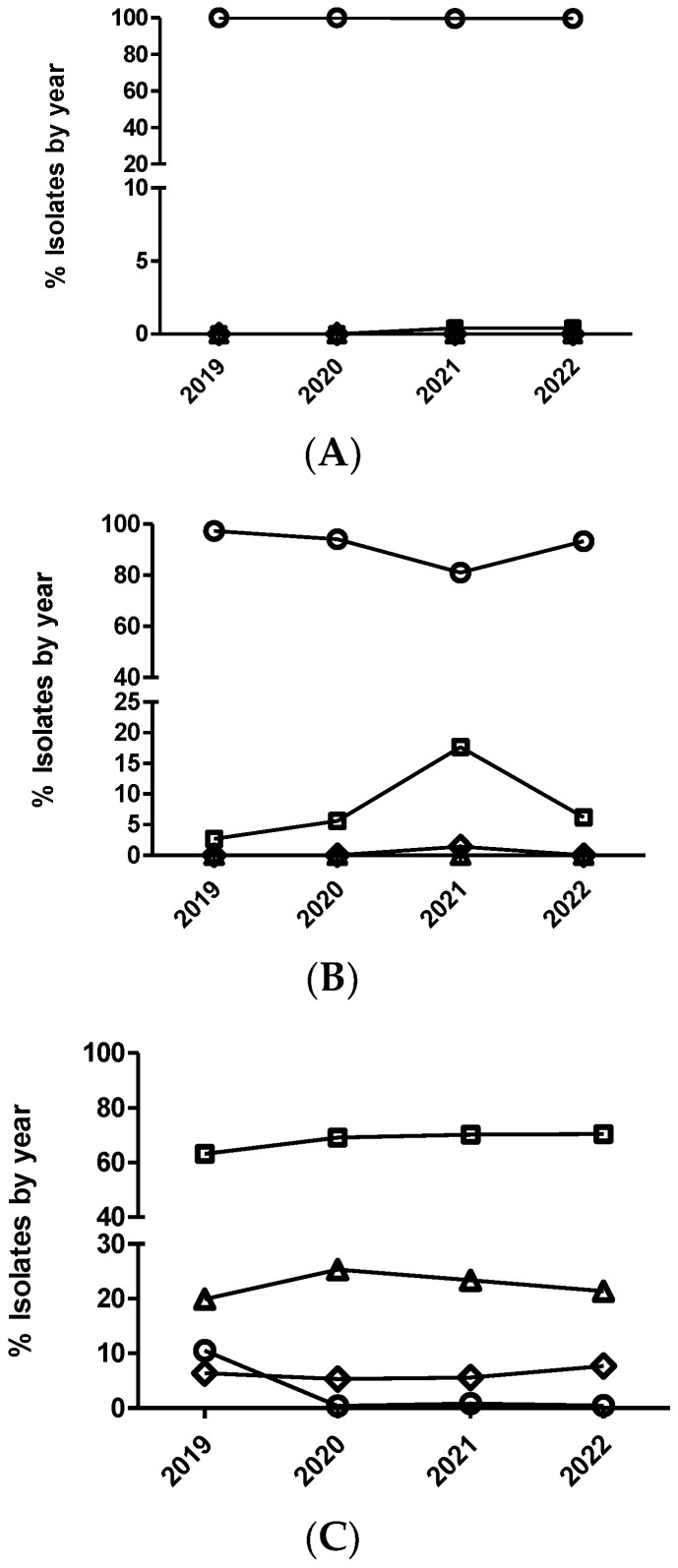
Percentage of *Actinobacillus pleuropneumoniae* (**A**), *Pasteurella multocida* (**B**), and *Escherichia coli* (**C**) isolates belonging to antimicrobial susceptibility category 1 (circle), category 2 (square), category 3 (diamond), and category 4 (triangle) for ceftiofur. The number of categories for multinominal analysis was based on distributing the range of MIC values in four categories (from one to four), equally distributed, that include two MIC values per category, with category one being the most susceptible (lowest MIC value) and category four the least susceptible (highest MIC value).

**Figure 5 antibiotics-12-01575-f005:**
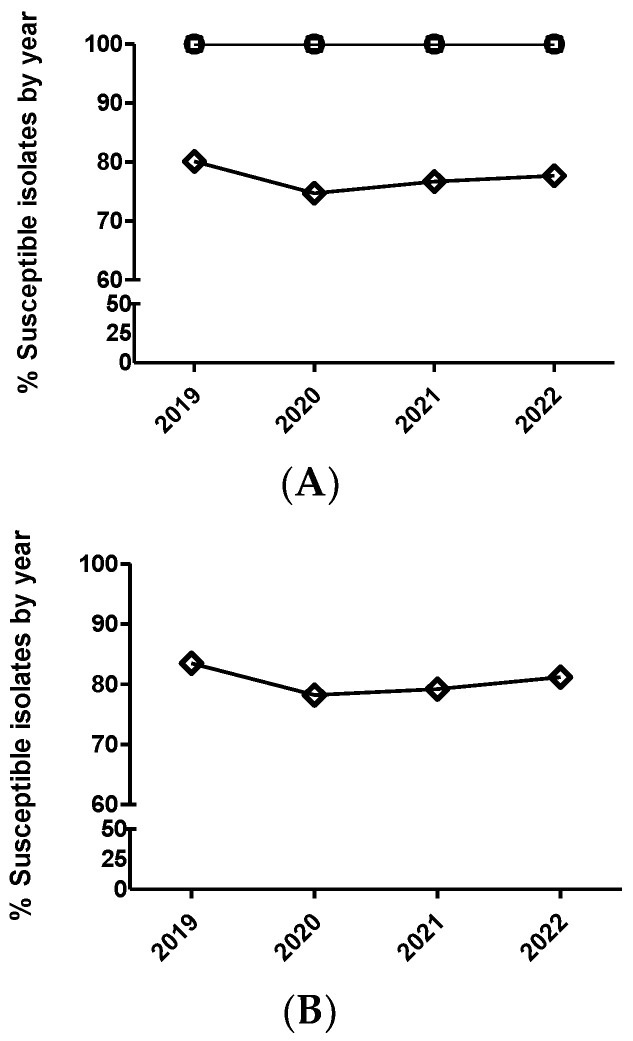
Percentage of susceptible isolates by year for ceftiofur (**A**) of *Actinobacillus pleuropneumoniae* (circle), *Pasteurella multocida* (square), and *Escherichia coli* (diamond), as well as cefquinome (**B**) of *Escherichia coli* (diamond), using the CLSI and EUCAST clinical breakpoints, as detailed in the Materials and Methods section.

**Figure 6 antibiotics-12-01575-f006:**
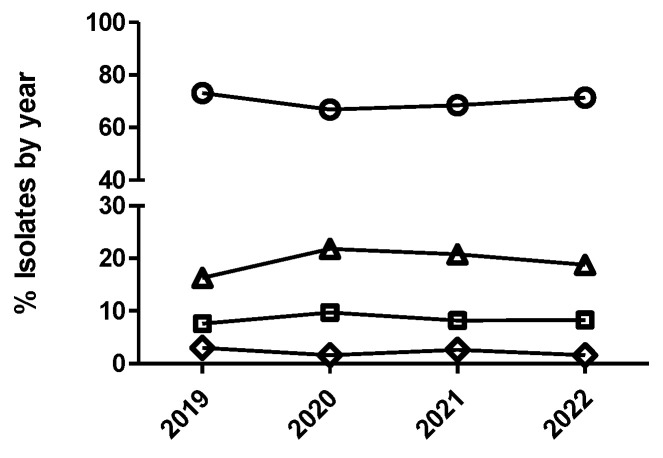
Percentage of *Escherichia coli* isolates belonging to antimicrobial susceptibility category 1 (open circle), category 2 (open squares), category 3 (open diamond), and category 4 (open triangle) for cefquinome. The number of categories for multinominal analysis was based on distributing the range of MIC values in four categories (from one to four), equally distributed, that include two MIC values per category, with category one being the most susceptible (lowest MIC value) and category four the least susceptible (highest MIC value).

**Figure 7 antibiotics-12-01575-f007:**
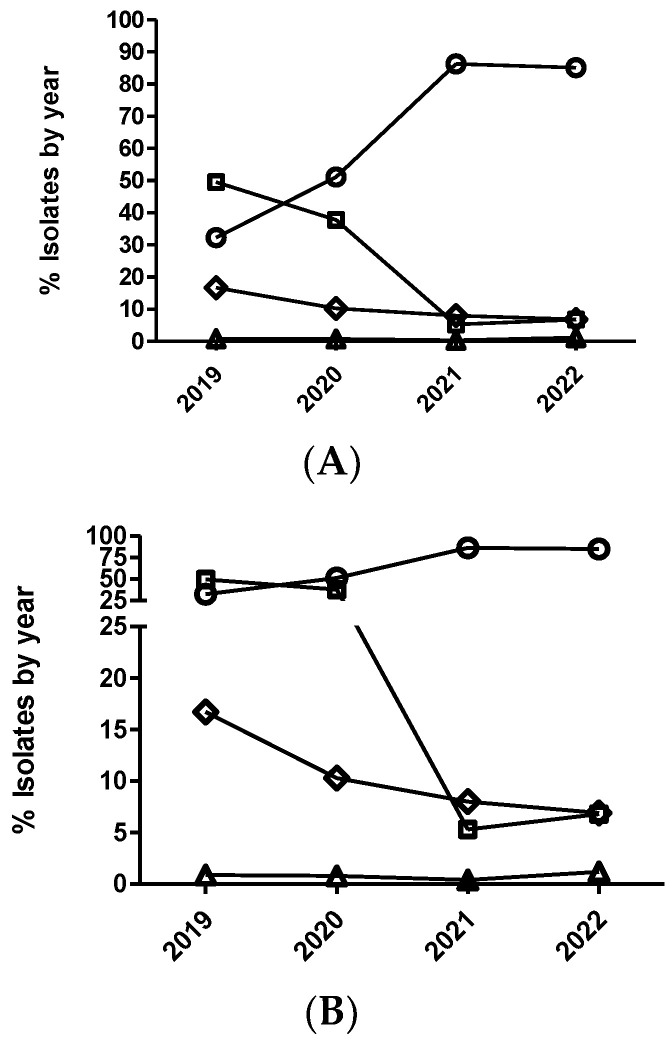
Percentage of *Escherichia coli* isolates belonging to antimicrobial susceptibility category 1 (circle), category 2 (square), category 3 (diamond), and category 4 (triangle) for colistin. The number of categories for multinominal analysis was based on distributing the range of MIC values in four categories (from one to four), equally distributed, that include two MIC values per category, with category one being the most susceptible (lowest MIC value) and category four the least susceptible (highest MIC value). It has been represented with two different scales (**A**,**B**) for the Y-axis to have more detail for the categories with low percentages of isolates.

**Figure 8 antibiotics-12-01575-f008:**
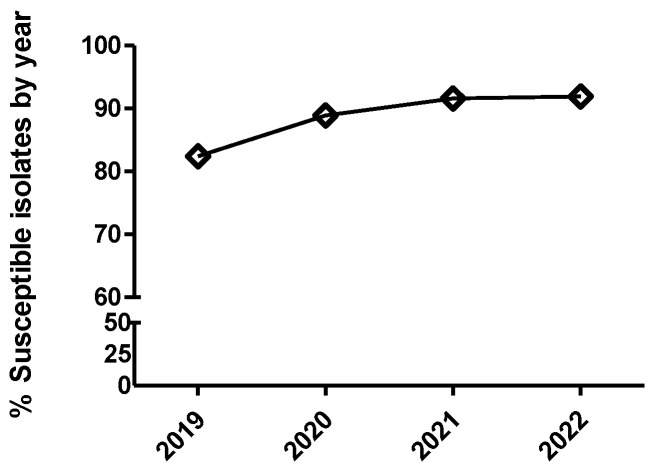
Percentage of susceptible isolates by year for colistin of *Escherichia coli* (diamond), using the CLSI and EUCAST clinical breakpoints, as detailed in the Materials and Methods section.

**Table 1 antibiotics-12-01575-t001:** Number of *Actinobacillus pleuropneumoniae* (APP), *Pasteurella multocida*, and *Escherichia coli* isolates isolated during the study period (2019–2022). The number of samples submitted per year is also provided in brackets to identify the suspected pathogen.

Pathogen	2019	2020	2021	2022
APP	123 (154)	195 (244)	237 (296)	228 (285)
*P. multocida*	111 (139)	100 (125)	147 (184)	178 (223)
*E. coli*	563 (710)	512 (645)	735 (927)	1082 (1364)

**Table 2 antibiotics-12-01575-t002:** Minimum inhibitory concentration (MIC) distribution values of (A) *Actinobacillus pleuropneumoniae*, (B) *Pasteurella multocida*, *and* (C) *Escherichia coli* to quinolones (enrofloxacin and marbofloxacin) from 2019 to 2022 in Spain. MIC range is the minimum and maximum MIC value observed. MIC90 and MIC50 values are the lowest concentration of the antibiotic at which 90 and 50% of the isolates were inhibited, respectively.

**A.** ***Actinobacillus pleuropneumoniae***			
*Enrofloxacin*			
**Year**	**MIC range**	**MIC_50_**	**MIC_90_**
2019	0.03–4	0.06	1
2020	0.03–4	0.06	0.5
2021	0.03–4	0.06	0.5
2022	0.03–4	0.06	0.5
*Marbofloxacin*			
**Year**	**MIC range**	**MIC_50_**	**MIC_90_**
2019	0.03–4	0.06	1
2020	0.03–4	0.03	0.5
2021	0.03–2	0.03	0.25
2022	0.03–4	0.03	0.25
**B.** ***Pasteurella multocida***			
*Enrofloxacin*			
**Year**	**MIC range**	**MIC_50_**	**MIC_90_**
2019	0.03–0.5	0.03	0.12
2020	0.03–0.5	0.03	0.12
2021	0.03–4	0.03	0.12
2022	0.03–4	0.03	0.12
*Marbofloxacin*			
**Year**	**MIC range**	**MIC_50_**	**MIC_90_**
2019	0.03–0.5	0.03	0.12
2020	0.03–0.5	0.03	0.12
2021	0.03–4	0.03	0.12
2022	0.03–4	0.03	0.12
**C.** ***Escherichia coli***			
*Enrofloxacin*			
**Year**	**MIC range**	**MIC_50_**	**MIC_90_**
2019	0.03–4	0.5	4
2020	0.03–4	1	4
2021	0.03–4	1	4
2022	0.03–4	0.5	4
*Marbofloxacin*			
**Year**	**MIC range**	**MIC_50_**	**MIC_90_**
2019	0.03–4	0.5	4
2020	0.03–4	0.5	4
2021	0.03–4	0.5	4
2022	0.03–4	0.5	4

**Table 3 antibiotics-12-01575-t003:** Minimum inhibitory concentration (MIC) distribution values of *Actinobacillus pleuropneumoniae*, *Pasteurella multocida*, and *Escherichia coli* to ceftiofur and cefquinome (only *Esherichia coli*) from 2019 to 2022 in Spain. MIC range is the minimum and maximum MIC value observed. MIC90 and MIC50 values are the lowest concentration of the antibiotic at which 90 and 50% of the isolates were inhibited, respectively.

*Actinobacillus pleuropneumoniae* and ceftiofur			
**Year**	**MIC range**	**MIC_50_**	**MIC_90_**
2019	0.06–0.12	0.06	0.06
2020	0.06–0.12	0.06	0.06
2021	0.06–0.25	0.06	0.06
2022	0.06–0.25	0.06	0.06
*Pasteurella multocida* and ceftiofur			
**Year**	**MIC range**	**MIC_50_**	**MIC_90_**
2019	0.06–0.25	0.06	0.12
2020	0.06–0.5	0.06	0.12
2021	0.06–1	0.06	0.25
2022	0.06–0.5	0.06	0.12
*Escherichia coli* and ceftiofur			
**Year**	**MIC range**	**MIC_50_**	**MIC_90_**
2019	0.06–8	0.5	8
2020	0.12–8	0.5	8
2021	0.12–8	0.5	8
2022	0.06–8	0.5	8
*Escherichia coli* and *cefquinome*			
**Year**	**MIC range**	**MIC_50_**	**MIC_90_**
2019	0.06–8	0.06	8
2020	0.06–8	0.12	8
2021	0.06–8	0.12	8
2022	0.06–8	0.12	8

**Table 4 antibiotics-12-01575-t004:** Minimum inhibitory concentration (MIC) distribution values of *Escherichia coli* to polymyxins (colistin) from 2019 to 2022 in Spain. MIC range is the minimum and maximum MIC value observed. MIC90 and MIC50 values are the lowest concentration of the antibiotic at which 90 and 50% of the isolates were inhibited, respectively.

Year	MIC Range	MIC_50_	MIC_90_
2019	0.5–32	1	1
2020	0.5–16	0.5	1
2021	0.5–16	0.5	0.5
2022	0.5–16	0.5	0.5

**Table 5 antibiotics-12-01575-t005:** The adjusted odds ratio (95% confidence interval) describing the annual variation in the susceptibility of *A. pleuropneumoniae* isolates to enrofloxacin using the logistic and multinominal regression model. The number of *A. pleuropneumoniae* isolates by year is detailed in Table 1.

	Logistic Analysis (Susceptible/Non-Susceptible)	Multinominal Analysis (MIC Outcome Categories Being Compared)		
Predictor Variable	NA	1 vs. 2	1 vs. 3	1 vs. 4
Year (2019–2022)	*p* = 0.0002	*p* = 0.002		
20 vs. 19	2.7 (1.6–4.8)	NS	2.1 (1.4–3.1)	2.3 (1.3–4.1)
21 vs. 20	NS	NS	NS	NS
22 vs. 21	NS	NS	NS	NS

NS means non-significant (*p* > 0.05), NA means not applicable.

**Table 6 antibiotics-12-01575-t006:** The adjusted odds ratio (95% confidence interval) describing the annual variation in the susceptibility of *A. pleuropneumoniae* isolates to marbofloxacin using the logistic and multinominal regression model. The number of *A. pleuropneumoniae* isolates by year is detailed in Table 1.

	Logistic Analysis (Susceptible/Non-Susceptible)	Multinominal Analysis (MIC Outcome Categories Being Compared)		
Predictor Variable	NA	1 vs. 2	1 vs. 3	1 vs. 4
Year (2019–2022)	*p* < 0.0001	*p* < 0.0001		
20 vs. 19	2.6 (1.5–4.6)	NS	3.2 (2.1–4.8)	2.2 (1.1–3.9)
21 vs. 20	NS	0.38 (0.18–0.69)	NS	NS
22 vs. 21	NS	NS	NS	NS

NS means non-significant (*p* > 0.05), NA means not applicable.

**Table 7 antibiotics-12-01575-t007:** The adjusted odds ratio (95% confidence interval) describing the annual variation in susceptibility of *E. coli* isolates to marbofloxacin using the logistic and multinominal regression model. The number of *E. coli* isolates by year is detailed in Table 1.

	Logistic Analysis (Susceptible/Non-Susceptible)	Multinominal Analysis (MIC Outcome Categories Being Compared)		
Predictor Variable	NA	1 vs. 2	1 vs. 3	1 vs. 4
Year (2019–2022)	NS	*p* < 0.0001		
20 vs. 19	NS	1.5 (1.2–1.9)	0.8 (0.7–0.9)	NS
21 vs. 20	NS	NS	NS	NS
22 vs. 21	NS	NS	NS	NS

NS means non-significant (*p* > 0.05), NA means not applicable.

**Table 8 antibiotics-12-01575-t008:** The adjusted odds ratio (95% confidence interval) describing the annual variation in the susceptibility of *E. coli* isolates to ceftiofur using the logistic and multinominal regression model. The number of *E. coli* isolates by year is detailed in Table 1.

	Logistic Analysis (Susceptible/Non-Susceptible)	Multinominal Analysis (MIC Outcome Categories Being Compared)		
Predictor Variable	NA	1 vs. 2	1 vs. 3	1 vs. 4
Year (2019–2022)	*p* = 0.15	*p* < 0.0001		
20 vs. 19	0.73 (0.55–0.98)	0.10 (0.06–0.16)	0.11(0.06–0.19)	0.10 (0.05–0.15)
21 vs. 20	NS	NS	NS	3.1 (1.2–12.4)
22 vs. 21	NS	NS	NS	NS

NS means non-significant (*p* > 0.05), NA means not applicable.

**Table 9 antibiotics-12-01575-t009:** The adjusted odds ratio (95% confidence interval) describing the annual variation in the susceptibility of *P. multocida* isolates to ceftiofur using the logistic and multinominal regression model. The number of *P. multocida* isolates by year is detailed in Table 1.

	Logistic Analysis (Susceptible/Resistant)	Multinominal Analysis (MIC Outcome Categories Being Compared)		
Predictor Variable	NA	1 vs. 2	1 vs. 3	1 vs. 4
Year (2019–2022)	NS		*p* = 0.0002	
20 vs. 19	NS	0.39 (0.13–0.88)	NS	NA
21 vs. 20	NS	NS	NS	NA
22 vs. 21	NS	3.1 (1.8–5.3)	NS	NA

NS means non-significant (*p* > 0.05), NA means not applicable. In this case, there are no isolates belonging to MIC category 4 (the least susceptible).

**Table 10 antibiotics-12-01575-t010:** The adjusted odds ratio (95% confidence interval) describing the annual variation in the susceptibility of *E. coli* isolates to colistin using the logistic and multinominal regression model. The number of *E. coli* isolates by year is detailed in Table 1.

	Logistic Analysis (Susceptible/Non-Susceptible)	Multinominal Analysis (MIC Outcome Categories Being Compared)		
Predictor Variable	NA	1 vs. 2	1 vs. 3	1 vs. 4
Year (2019–2022)	*p* < 0.0001	*p* < 0.0001		
20 vs. 19	1.7 (1.2–2.4)	5.5 (4.6–6.9)	3 (2.4–3.8)	NS
21 vs. 20	NS	2.7 (2.3–3.3)	NS	NS
22 vs. 21	NS	0.23 (0.17–0.29)	0.55 (0.44–0.70)	0.37 (0.12–0.86)

NS means non-significant (*p* > 0.05), NA means not applicable.

**Table 11 antibiotics-12-01575-t011:** Clinical breakpoints (susceptible/non-susceptible for each isolate) used according to the CLSI clinical breakpoints for *Actinobacillus pleuropneumoniae* (APP), *Pasteurella multocida* (PM), and *Escherichia coli* (EC) for quinolones and cephalosporins, and the EUCAST guidelines for colistin in the case of *E. coli*, respectively.

Antimicrobial	APP		PM		EC	
	Susceptible	Non-Susceptible	Susceptible	Non-Susceptible	Susceptible	Non-Susceptible
Enrofloxacin	<0.25	>0.25	<0.25	>0.25	<0.25 *	>0.25 *
Marbofloxacin	<0.25	>0.25	<0.25	>0.25	<0.25 *	>0.25 *
Ceftiofur	<2	>2	<2	>2	<2 *	>2 *
Cefquinome	NA	NA	NA	NA	<2 *	>2 *
Colistin	NA	NA	NA	NA	<2	>2

NA—Not applicable for this study, * extrapolated from respiratory to digestive pathogens.

## Data Availability

The data presented in this study are available on reasonable request from the corresponding author. The data are not publicly available due to confidentiality issues related with clinical cases.

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
