# Peer review of "The Susceptibility Trends of Respiratory and Enteric Porcine Pathogens to Last-Resource Antimicrobials"

_antibiotics, 2023, doi:10.3390/antibiotics12111575_

Round 1
Reviewer 1 Report
Comments and Suggestions for Authors
The manuscript entitled “Susceptibility trends of respiratory and enteric porcine pathogens to last resource antimicrobials” reported the AMR bacteria isolated from porcine with PRDC and PWD during observation between 2019 to 2022. The MS can not be accepted in the current version because the presentation of overall results is not good and makes a confusing for readers, they need more improvement. This MS should be written as per journal format and style. Some bacteria were also included, but not completed in the experiments. I suggest that these should be excluded from this MS. By the way, there are major flaws found and must be extensively revised as follows:
1. The short background of this study should be included in the Abstract section before the objectives.
2. Line 110: As you mentioned, 1,837 samples from PRDC and 3,813 samples from PWD. Kindly add the table presenting the number of samples collected each year and add the number of positive samples and the % prevalence of bacteria found in each sample per year. That was a trend!!!
3. It would be better if this MS only reported on APP, P. multocida, and E. coli because the overall results were completed on the above bacteria. However, this depends on the author's decision. Don’t forget to remove the supplementary file.
4. Tables 2-5 should be revised, they are more ambiguous and need to follow the journal formatting.
5. Tables 2-5: the values MIC range, MIC50, and MIC90 should be clarified in the footnote, they are from the average, median, or mode values of N isolates.
6. Tables 2-5 can combine as one big Table with more columns, the current version is not a good presentation and is unable to overview the trend. If some drugs were not performed on some bacteria, you can blank the column and provide the reason in the footnote.
7. Figures 1-7: need to be improved as per journal formatting. The order of figures should be rearranged as APP, P.multocida, and E. coli and present on the same page. The figure caption should also be improved.
8. Figures 1-7: category 1 to 4 should be directly defined. For example, a circle line represents the lowest MIC of the drug, triangle line represents ... or something. This is ambiguous for the reader!!!!! OR the information of each category must be provided as a footnote.
9. Table 6-11: Add full year “2019 VS 2022”. What is the meaning of 1, 2, 3, and 4? These should be provided in the footnote.
10. Why the samples with PWD were only collected from humanely euthanized animals? In my opinion, the samples can be collected from live pigs with diseases such as rectal swabs and/or fecal content. Animals can continually survive and will be taken care of by the farm owners. I think it would be better if this matter should be in your consideration in future works for preventing the losses of the economic situation and production yield on the pig farmers (3,813 samples = 3,813 pigs?).
Good Luck!!!
Comments on the Quality of English LanguageCheck some words and phrases with grammar errors throughout the document.
Author Response
Reviewer 1:
The manuscript entitled “Susceptibility trends of respiratory and enteric porcine pathogens to last resource antimicrobials” reported the AMR bacteria isolated from porcine with PRDC and PWD during observation between 2019 to 2022. The MS can not be accepted in the current version because the presentation of overall results is not good and makes a confusing for readers, they need more improvement. This MS should be written as per journal format and style. Some bacteria were also included, but not completed in the experiments. I suggest that these should be excluded from this MS. By the way, there are major flaws found and must be extensively revised as follows:
- The short background of this study should be included in the Abstract section before the objectives.
I would like to include a short background in the abstract section but there are strict rules in relation with the number of words and our abstract has the maximum extension accepted for the journal. We believe that the interest and the reason to carry out this kind of studies is clear in this journal.
- Line 110: As you mentioned, 1,837 samples from PRDC and 3,813 samples from PWD. Kindly add the table presenting the number of samples collected each year and add the number of positive samples and the % prevalence of bacteria found in each sample per year. That was a trend!!!
Thanks a lot for this comment. This information has been added in Table 1 as suggested by the referee and it has been corrected due to this recommendation because some cases was erroneously counted twice.
- It would be better if this MS only reported on APP, multocida, and E. colibecause the overall results were completed on the above bacteria. However, this depends on the author's decision. Don’t forget to remove the supplementary file.
In Table 1, we have only maintained APP, P Multocida and E coli because the study was carried out with them. Nevertheless, brief information for other pathogens has been included in the main text.
We prefer to maintain the supplementary file. It is true that the number of isolates is quite low for Salmonella spp but it supports the results obtained for Escherichia coli and it deserves to be included as supplementary information in our opinion.
- Tables 2-5 should be revised, they are more ambiguous and need to follow the journal formatting.
We have revised the tables according to this referee and other referee´s comments to be consistent with all the revision.
- Tables 2-5: the values MIC range, MIC50, and MIC90 should be clarified in the footnote, they are from the average, median, ormode values of N isolates.
MIC range, MIC50 and MIC90 are the usual way to show MIC distribution data. In any case, it has been added a clarification as suggested by the referee.
- Tables 2-5 can combine as one big Table with more columns, the current version is not a good presentation and is unable to overview the trend. If some drugs were not performed on some bacteria, you can blank the column and provide the reason in the footnote.
It is not possible to represent all the information in one single table. As suggested by the referee, this table should include 33 columns if table 2-5 is combined. As suggested by other referee, we have included information belonging to different antimicrobial families in three tables (2-4): quinolones, cephalosporins and polymixins.
- Figures 1-7: need to be improved as per journal formatting. The order of figures should be rearranged as APP, multocida,and E. coli and present on the same page. The figure caption should also be improved.
Figure caption has been improved as suggested by the referee. In relation with formatting, I am sure that, if the paper is accepted, the journal editors will help me with the figures to fix them in one page as suggested by the referee.
- Figures 1-7: category 1 to 4 should be directly defined. For example, a circle line represents the lowest MIC of the drug, triangle line represents ... or something. This is ambiguous for the reader!!!!! OR the information of each category must be provided as a footnote.
It has been included as suggested by the referee.
- Table 6-11: Add full year “2019 VS 2022”. What is the meaning of 1, 2, 3, and 4? These should be provided in the footnote.
It has been included as suggested by the referee.
- Why the samples with PWD were only collected from humanely euthanized animals? In my opinion, the samples can be collected from live pigs with diseases such as rectal swabs and/or fecal content. Animals can continually survive and will be taken care of by the farm owners. I think it would be better if this matter should be in your consideration in future works for preventing the losses of the economic situation and production yield on the pig farmers (3,813 samples = 3,813 pigs?).
This phrase must have been misunderstood by the referee. Most of the samples came from sick animals (rectal swabs). Only severely diseased animals were sacrificed by humanitarian reasons. As commented by the referee, it does not make any sense to sacrifice animals with this goal. This point has been clarified in the revised version.
Reviewer 2 Report
Comments and Suggestions for Authors
See attached file

See attached file
Author Response
Reviewer 2
Susceptibility trends of respiratory and enteric porcine pathogens to last resource antimicrobials
Reviewer´s comments are answered point by point in bold letter.
Line # |
Comment |
Abstract |
The results are well presented. 1-2 lines may be added discussing the importance of results, followed by a conclusion (already included).
Thanks a lot for your comment in relation with the presentation of results. It is really appreciated. There is a limitation in the extension of the abstract. It cannot be widened more. The importance of the results is detailed in the discussion section.
|
Line-39-45 |
Other possible sources of AMR may be mentioned here (line-43), as misuse of drugs in not the only reason for AMR.
A comment has been included to clarify this phrase because, as commented by the referee, the misuse of drugs is not the only reason for AMR.
As we are discussing AMR importance in animal and human, more recent reference may be mentioned here including: · https://doi.org/10.3390/antibiotics11081082 Antimicrobial Resistance and Its Spread Is a Global Threat · https://doi.org/10.3390/ejihpe11010006 Antimicrobial Resistance in the Context of the Sustainable Development Goals: A Brief Review · http://pvj.com.pk/pdf-files/42_2/167-172.pdf Trends in Frequency, Potential Risks and Antibiogram of E. coli Isolated from Semi-Intensive Dairy Systems · http://pvj.com.pk/pdf-files/41_4/519-523.pdf Antimicrobial Resistance, Adhesin and Toxin Genes of Porcine Pathogenic Escherichia coli Following the Ban on Antibiotics as the Growth Promoters in Feed. · http://dx.doi.org/10.29261/pakvetj/2022.074 Molecular Detection of Biofilm Production among Multidrug Resistant Isolates of Pseudomonas aeruginosa from Meat Samples
Thanks a lot for this comment. Two additional references (new ones) have been included as suggested by the reviewer.
|
Line-52-55 |
The sentence seems too long. It may be separated into two sentences for better understanding of the reader.
Thanks a lot for this comment. The sentence has been split in two phrases for improving the understanding of the reader.
For colistin resistance, it may be helpful to include the following references to highlight the importance in recent years.
· http://dx.doi.org/10.29261/pakvetj/2020.077 Detection of Colistin Resistance in Mannheimia haemolytica & Pasteurella multocida Isolates from Ruminants in Morocco · http://dx.doi.org/10.29261/pakvetj/2021.016 Molecular Detection of Colistin Resistance Gene (MCR-1) in E. coli Isolated from Cloacal Swabs of Broilers
One reference has been included as suggested by the referee.
|
The concept of ONE health may be mentioned here, as we are discussing the human-animal-environment interface. A comment has been included as suggested by the referee.
|
|
The full form of AMU may please be mentioned before using the abbreviation.
It has been mentioned the abbreviation of AMU as suggested by the referee.
|
|
The following recent reference may be cited for AMU specifically regarding colistin resistance. · https://doi.org/10.1016/S2666-5247(22)00387-1 International manufacturing and trade in colistin, its implications in colistin resistance and One Health global policies: a microbiological, economic, and anthropological study
This reference has been included as suggested by the referee.
|
Line-76 |
“…… veterinary antimicrobial use”
It has been corrected as suggested by the referee.
|
Line-83 |
“Moreover, Actinobacillus………….
It has been corrected as suggested by the referee.
|
Line-91 |
[3-32], Was it meant to be like? [3, 32]
Thanks for this comment. It has been clarified according to referee´s suggestion.
|
Line-110 |
“from isowean” ………animal (sow)???
It has been corrected as suggested by the referee.
|
Line-113 |
“Clinical signs” instead of clinical sings
It has been corrected as suggested by the referee.
|
Line-120 Line-503 |
“Finally, in 5% of the enteric samples, it was possible to isolate more than one bacterial species, generally Salmonella spp.”
It means only Salmonella species were isolated along other bacteria??
Yes, in our case, Salmonella spp used isolated in clinical cases besides Escherichia coli. It is not necessary to correct it according to our understanding.
|
Line-68 Line-121 Line-609 |
“Salmonella spp” may be replaced as “Salmonella species” Same in the Table 1 and Table 1 heading-line-126
it has been corrected as suggested by the referee.
|
Bacteria isolation |
Only total percentage of bacterial isolates are mentioned, however, percentage of each bacterial isolated may be mentioned.
As suggested by other referees, Table 1 has ben updated to cope with referee´s suggestion in this point.
|
Table-1 |
Number of isolates from PRDC and PWD may be separately described as the total number of samples are different in both cases
it has been corrected as suggested by the referee.
|
Percentage of each isolated bacteria species may be mentioned in parentheses like 123 (--%), as number of total samples are different for different bacterial isolates. |
|
B. bronchiseptica “B.” may be italicized
This bacteria has been deleted due to other referee´s suggestion but a comment still remains in the paper.
|
|
Tables 2-4 |
A similar formatting type may be used for better understanding. Like Table 2 elaborates MIC of all bacteria to quinolones Same formatting may be used in depicting MIC of all bacteria to cephalosporins (ceftiofur & cefquinome) (as Table 3)
Thanks a lot for this comment. It has been corrected as suggested by the referee and referee 1 makes a similar suggestion.
|
Line-197 Line-228 Line-283 Line-290 348 373 381 420 |
“not significant” to non-significant
it has been corrected as suggested by the referee.
|
Line-243 |
“Interestingly, it was not observed any significant trend using dichotomized data for this drug/microorganism combination.” May be rephrased for better understanding
it has been rephrased as suggested by the referee.
|
Line-293 |
3rd and 4th cephalosporins. “3rd and 4th generation cephalosporins.” It has been corrected as suggested by the referee.
|
Figure-5 |
It may be mentioned ceftiofur and cefquinome were used against which bacteria, as figure-5 (B) shows that cefquinome was only used for E. coli
Thanks a lot for this comment. It has been corrected as suggested by the referee.
|
Line-440 |
EARS-VET Full form may be mentioned first time It has been corrected as suggested by the referee.
|
Line-470-473 |
Why testing one of those fluroquinolones is sufficient? It may be discussed here? In this paragraph, this matter has been thoroughly discussed. We consider unnecessary to extend it more.
|
Line-482 |
Antibiotic resistance or antimicrobial resistance?
It has been corrected as suggested by the referee.
|
Line 496 |
Spain from 34.9 mg/PCU to 3 mg/PCU between 2015 and 2018, Please cite the respective reference here
It has been included as suggested by the referee.
|
Line-484-505 |
Only E. coli is discussed. A. pleuropneumoniae, P. multocida???
It has been corrected according to referee´s suggestion.
|
Clinical samples |
Number of samples may be mentioned here, emphasizing the size of the study
Thanks a lot for this comment. The number of samples has been included in this section as suggested by the referee.
|
Line-533 Line-559 |
5-10% CO2 5-10% CO2
it has been corrected as suggested by the referee.
|
Line-562 Line-590 |
Italicize “E. coli” it has been corrected as suggested by the referee.
|
Line-586 |
0,06-8, 0,03-4, 0,25-32 You mean 0.06 etc.?
Yes, it has been corrected as suggested by the referee.
|
Reviewer 3 Report
Comments and Suggestions for Authors
In this study, the authors isolated porcine bacteria and analyzed the antimicrobial susceptibility trends of these bacteria to quinolones, cephalosporins, and polymyxins. The trend of drug susceptibility from 2019 to 2022 was analyzed based on dichotomized or categorized data. The authors found that categorized data can more sensitively detect the trend of AMR profile change than dichotomized data, which is very interesting and can provide some implications for the control of drug use. One thing that requires improvement is the explanation of methods. It was not clear how many farms were included in the study, and it was also unclear whether including only one isolate per farm would result in an under-representation of bacterial clones. Another concern is whether data from 4 years is sufficient to support their conclusion. The authors should also appropriately use italics type of bacteria name (E. coli ) and change “CO2” to “CO2” throughout the paper. Please see my comments below:
L54: Explain AMU.
L113-116: To prevent redundancy and overrepresentation of bacterial clones, the authors only included one isolate from each farm. What are the chances that the bacterial clones were under-represented by including only one isolate from each farm? Does this mean from each farm, only one isolate was collected each year, or four years? Did the authors perform genomic analyses to confirm this, or was there any literature to support the authors’ statement?
It was unclear whether this means one isolate per species, or one isolate regardless of species were collected from each farm. It was also not mentioned from how many farms the samples were collected and how many samples were collected from each farm. It would be more clear if the authors could introduce how many samples were collected from each farm and how many isolates were collected for each species from each farm.
Table 6: Does the range mean confidence intervals? Please specify.
Figure 5 panel B: Was cefquinome only tested for E. coli? Please specify in the text.
L384-386: all categories are labeled as “open circle/squares etc”. These descriptions are inconsistent with the previous figures.
L582-586: the authors described the number of categories was decided by dividing the range of MIC values into four groups, each with two MIC values. Does this mean the range was divided into four parts equally? How do these categories correlate with the clinical breakpoints? It would be clearer to introduce this briefly.
L506-509: I agree that the results have some implications on the control of antimicrobial use to minimize the spread of resistance. However, the authors should carefully evaluate whether these results are sufficiently sound to support the conclusion since the authors only analyzed data from 4 years. What is the average time for these bacteria to acquire resistance to these antimicrobials? How fast would bacteria lose resistance when the antimicrobial is removed? Were these trends reported anywhere else in the world?
All figures: The authors used “strains” rather than “isolates” in all y-axis, which is not consistent with the statement in their main text. Different isolates may come from the same strain. I suggest changing “strains” to “isolates” in their main text unless the authors confirm that each isolate represents one distinct strain in this study and thus can be interchangeably used.
Author Response
Reviewer 3:
In this study, the authors isolated porcine bacteria and analyzed the antimicrobial susceptibility trends of these bacteria to quinolones, cephalosporins, and polymyxins. The trend of drug susceptibility from 2019 to 2022 was analyzed based on dichotomized or categorized data. The authors found that categorized data can more sensitively detect the trend of AMR profile change than dichotomized data, which is very interesting and can provide some implications for the control of drug use.
Thank you very much for this comment.
One thing that requires improvement is the explanation of methods. It was not clear how many farms were included in the study, and it was also unclear whether including only one isolate per farm would result in an under-representation of bacterial clones. Another concern is whether data from 4 years is sufficient to support their conclusion.
Authors have explained in more detail the number of farms included in the study. In relation with the rest of questions are addressed below.
The authors should also appropriately use italics type of bacteria name (E. coli ) and change “CO2” to “CO2” throughout the paper.
it has been corrected as suggested by the referee.
Please see my comments below:
L54: Explain AMU.
It has been corrected as also suggested by other referees.
L113-116: To prevent redundancy and overrepresentation of bacterial clones, the authors only included one isolate from each farm. What are the chances that the bacterial clones were under-represented by including only one isolate from each farm? Does this mean from each farm, only one isolate was collected each year, or four years? Did the authors perform genomic analyses to confirm this, or was there any literature to support the authors’ statement?
One key question to determine prevalence is epidemiological studies is that samples must be randomly selected from the population. In the case of dichotomized data, the goal is to determine a prevalence (%susceptible or non-susceptible isolates). In our case, samples were randomly selected from the database every year but excluding this farm for following years. This procedure allows determining prevalence in a robust way every year and allows studying the antimicrobial susceptibility trend in the population because samples are independent.
It was unclear whether this means one isolate per species, or one isolate regardless of species were collected from each farm. It was also not mentioned from how many farms the samples were collected and how many samples were collected from each farm. It would be more clear if the authors could introduce how many samples were collected from each farm and how many isolates were collected for each species from each farm.
This point has been included in Table 1 as suggested by this and other referees.
Table 6: Does the range mean confidence intervals? Please specify.
In Table 6, it is detailed the following:
Table 6.- The adjusted odds ratio (95% confidence interval) describing the annual variation in susceptibility of A. pleuropneumoniae isolates to marbofloxacin using the logistic and multinominal regression model. The number of A. pleuropneumoniae isolates by year is detailed in Table 1.
It is clearly detailed the ratio and 95% confidence interval. We do not believe that additional information must be added.
Figure 5 panel B: Was cefquinome only tested for E. coli? Please specify in the text.
It has been clarified in the revised version of the paper.
L384-386: all categories are labeled as “open circle/squares etc”. These descriptions are inconsistent with the previous figures.
We do not understand this comment. We have used the same labelling for multinominal studies.
L582-586: the authors described the number of categories was decided by dividing the range of MIC values into four groups, each with two MIC values. Does this mean the range was divided into four parts equally? How do these categories correlate with the clinical breakpoints? It would be clearer to introduce this briefly.
The range was divided into four parts equally. This point has been clarified in the revised version of the paper.
Our analysis using susceptible/non-susceptible and categorized MIC data (MIC category 1, 2, 3 and 4) were used for logistic and multinominal logistic regression model, respectively as dependent variables, and the year as independent one. Thus, they are two totally independent system of analysis. It is not necessary to correlate the MIC categories 1, 2, 3 and 4 with the clinical breakpoints because they are totally independent between each other.
L506-509: I agree that the results have some implications on the control of antimicrobial use to minimize the spread of resistance. However, the authors should carefully evaluate whether these results are sufficiently sound to support the conclusion since the authors only analyzed data from 4 years. What is the average time for these bacteria to acquire resistance to these antimicrobials? How fast would bacteria lose resistance when the antimicrobial is removed? Were these trends reported anywhere else in the world?
Authors agree with referee that four years is a quite short period of time to analyse the susceptibility trends of respiratory and enteric pathogens to last resource antimicrobials, but we are using a robust statistical methodology that allows comparison between the whole period and year of sampling was categorized by individual years and modelled as a hierarchical indicator variable, where for each year the preceding year was used as the referent. Thus, we are confident with the results obtained. In any case, we have added this important point in the discussion as a limitation of the study. It is not known the average time for these bacteria to acquire resistance to these antimicrobials and how fast these bacteria would lose resistance when the antimicrobial is removed. Thus, this interesting point cannot be included in the discussion section.
All figures: The authors used “strains” rather than “isolates” in all y-axis, which is not consistent with the statement in their main text. Different isolates may come from the same strain. I suggest changing “strains” to “isolates” in their main text unless the authors confirm that each isolate represents one distinct strain in this study and thus can be interchangeably used.
It has been corrected as suggested by the referee.
Round 2
Reviewer 1 Report
Comments and Suggestions for Authors
The authors did not respond to my requests. This MS has not yet improved and is ambiguous. Journal formatting can be used to evaluate your attention to revise your work. The quality is not up to the standard of Antibiotics.
Comments on the Quality of English LanguageNo comment.
Author Response
Reviewer 1:
Round 2:
The authors did not respond to my requests. This MS has not yet improved and is ambiguous. Journal formatting can be used to evaluate your attention to revise your work. The quality is not up to the standard of Antibiotics.
Dear Reviewer,
I apologize for not considering all your suggestions to review our paper in this first round. We tried to do it in the first version, but we had to carry out it considering other reviewer´s suggestions that were contradictory in some way. In this round, we have addressed most of the recommendations. We really wish that this time it could be acceptable for publication.
Comments to round 1 and 2 together for referee 1:
The manuscript entitled “Susceptibility trends of respiratory and enteric porcine pathogens to last resource antimicrobials” reported the AMR bacteria isolated from porcine with PRDC and PWD during observation between 2019 to 2022. The MS can not be accepted in the current version because the presentation of overall results is not good and makes a confusing for readers, they need more improvement. This MS should be written as per journal format and style. Some bacteria were also included, but not completed in the experiments. I suggest that these should be excluded from this MS. By the way, there are major flaws found and must be extensively revised as follows:
- The short background of this study should be included in the Abstract section before the objectives.
We have included a short background in the abstract section in this second round. It is highlighted in red.
- Line 110: As you mentioned, 1,837 samples from PRDC and 3,813 samples from PWD. Kindly add the table presenting the number of samples collected each year and add the number of positive samples and the % prevalence of bacteria found in each sample per year. That was a trend!!!
Thanks a lot for this comment. In the first round of revision, this information had been added in Table 1 as suggested by the referee and it has been corrected due to this recommendation because some cases were erroneously counted twice. It is highlighted in red.
- It would be better if this MS only reported on APP, multocida, and E. colibecause the overall results were completed on the above bacteria. However, this depends on the author's decision. Don’t forget to remove the supplementary file.
In the first round of revision, we have only maintained APP, P Multocida and E coli in Table 1 because the study was carried out with them. Nevertheless, brief information for other pathogens has been included in the main text. It is highlighted in red.
The supplementary figure has been deleted as suggested by the referee in this round 2.
- Tables 2-5 should be revised, they are more ambiguous and need to follow the journal formatting.
In round one, we had revised the tables according to this referee and other referee´s comments to be consistent with all the revision. We believe that it follows journal format for tables.
- Tables 2-5: the values MIC range, MIC50, and MIC90 should be clarified in the footnote, they are from the average, median, ormode values of N isolates.
MIC range, MIC50 and MIC90 are the usual way to show MIC distribution data. In any case, it has been added a clarification as suggested by the referee in the first round of revision. It is highlighted in red.
- Tables 2-5 can combine as one big Table with more columns, the current version is not a good presentation and is unable to overview the trend. If some drugs were not performed on some bacteria, you can blank the column and provide the reason in the footnote.
Sorry to say that it is not possible to represent all the information in one single table. As suggested by other referee, we have included information belonging to different antimicrobial families in three tables (2-4): quinolones, cephalosporins and polymixins. Any additional change could be contradictory with other referee´s suggestions. We kindly ask to the editor to revise this point in detail according to round 1 of revision.
- Figures 1-7: need to be improved as per journal formatting. The order of figures should be rearranged as APP, multocida,and E. coli and present on the same page. The figure caption should also be improved.
Figure caption had been improved as suggested by the referee in the first round. We have corrected the order of the figures as suggested by the referee and the format has been updated to include one figure by page in second round of revision. It is highlighted in red.
- Figures 1-7: category 1 to 4 should be directly defined. For example, a circle line represents the lowest MIC of the drug, triangle line represents ... or something. This is ambiguous for the reader!!!!! OR the information of each category must be provided as a footnote.
It had been included as suggested by the referee in the first round of revision. It is highlighted in red.
- Table 6-11: Add full year “2019 VS 2022”. What is the meaning of 1, 2, 3, and 4? These should be provided in the footnote.
It had been included as suggested by the referee in the first round of revision. It is highlighted in red.
- Why the samples with PWD were only collected from humanely euthanized animals? In my opinion, the samples can be collected from live pigs with diseases such as rectal swabs and/or fecal content. Animals can continually survive and will be taken care of by the farm owners. I think it would be better if this matter should be in your consideration in future works for preventing the losses of the economic situation and production yield on the pig farmers (3,813 samples = 3,813 pigs?).
This phrase must have been misunderstood by the referee. Most of the samples came from sick animals (rectal swabs). Only severely diseased animals were sacrificed by humanitarian reasons. As commented by the referee, it does not make any sense to sacrifice animals with this goal. This point had been clarified in the revised version after first round. It is highlighted in red.
Round 3
Reviewer 1 Report
Comments and Suggestions for Authors
I have no request from the authors.
Comments on the Quality of English LanguageNo comment.